# Bootstrapping Vision-Language Learning with Decoupled Language Pre-training

**Yiren Jian**[1]    **Chongyang Gao**[2]    **Soroush Vosoughi**[1]
[1]Dartmouth College    [2]Northwestern University

## Abstract

We present a novel methodology aimed at optimizing the application of frozen large language models (LLMs) for resource-intensive vision-language (VL) pre-training. The current paradigm uses visual features as prompts to guide language models, with a focus on determining the most relevant visual features for corresponding text. Our approach diverges by concentrating on the language component, specifically identifying the optimal prompts to align with visual features. We introduce the Prompt-Transformer (P-Former), a model that predicts these ideal prompts, which is trained exclusively on linguistic data, bypassing the need for image-text pairings. This strategy subtly bifurcates the end-to-end VL training process into an additional, separate stage. Our experiments reveal that our framework significantly enhances the performance of a robust image-to-text baseline (BLIP-2), and effectively narrows the performance gap between models trained with either 4M or 129M image-text pairs. Importantly, our framework is modality-agnostic and flexible in terms of architectural design, as validated by its successful application in a video learning task using varied base modules. The code will be made available at `https://github.com/yiren-jian/BLIText`.

## 1 Introduction

The field of vision-language (VL) learning seeks to create AI systems that mimic human cognition, processing the world through multi-modal inputs. Core research areas in VL include visual-question-answering (VQA), image captioning, image-text retrieval, and visual reasoning. VL learning began with task-specific learning [3, 64] and has since progressed to large-scale image-text pre-training paired with task-specific fine-tuning [50]. Furthermore, contemporary studies have begun exploring the use of off-the-shelf frozen pre-trained large language models (LLMs) in VL models [2, 23, 34, 58], which have delivered impressive results in language generation tasks such as VQA and image captioning.

Present VL models utilizing frozen LLMs are characterized by shared design elements: visual encoders, visual-to-language modules, and frozen LLMs. Except for Flamingo [2], which employs a visual signal at each layer of the frozen LLM via gated cross-attention, the majority of works [6, 34, 41, 46, 58] feed aligned visual features as soft language prompts [29] into the frozen LLMs (see Figure 1 *left*). The models are then trained end-to-end with an image-conditioned language generation loss using large-scale image-text pairs. This conceptually simple and implementation-wise straightforward design has proven effective. BLIP-2 [34] demonstrates that decoupling the end-to-end training into two stages is crucial for state-of-the-art results. The second stage of training involves standard end-to-end learning, while the first stage of training of BLIP-2 utilizes a learnable module (called Query-Transformer/Q-Former) to selectively choose/query visual features relevant to the corresponding text. This reduces 256 features of an entire image to the 32 most relevant visual features that will be sent into the following parts of the model. Stage 1 of BLIP-2 can be viewed as a refined learnable version of early VL works [3, 38, 71] that use object detectors like Faster-RCNN

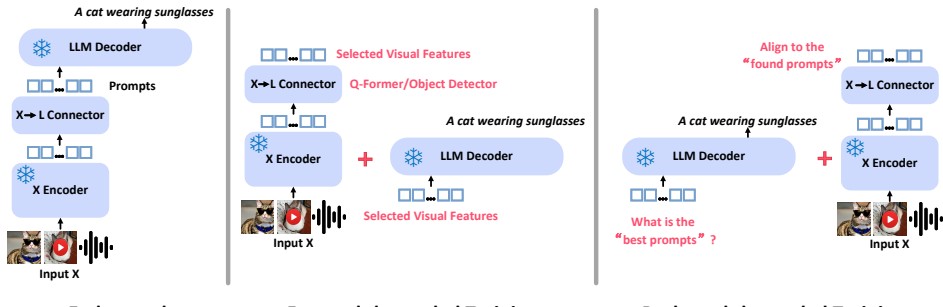

Figure 1: *left:* End-to-end training of X-to-language models (where X can be images, videos, or audio), in which aligned input features are provided as prompts to LLMs. Examples include Frozen [29] and ClipCap [46]. *middle:* "Forward-decoupled training" as demonstrated in BLIP-2 [34] and X-LLM [6]. For instance, in BLIP-2, the Q-Former is first trained to extract relevant features from the image encoder, and then the selected features are used as prompts for LLM for end-to-end learning. *right:* We propose "backward-decoupled training", which initially identifies the "reference prompt" for the LLM to generate the target text, followed by mapping input features to the "reference prompt".

[17] to select features from regions of objects (objects in images are likely to be mentioned and thus relevant to the accompanying text). We refer to this strategy as "forward-decoupling" since it uses a heuristic to learn/select which useful features are forward-passed into the subsequent model to mitigate challenges in the end-to-end optimization (shown in Figure 1 *middle*).

We provide a novel insight to mitigate the challenges in end-to-end optimization by introducing "backward-decoupling" during back-propagation. For a caption $t$ (e.g., *"a cat wearing sunglasses"*) from VL pre-training dataset $\mathcal{D}_{\text{VL}}$, the optimizer first finds the optimal continuous prompt $p$ for a fixed decoder LLM $D_{\text{language}}$: $p = \arg\min_p \mathcal{L}(D_{\text{language}}(p), t)$, before further back-propagating into the vision-to-language module (e.g., Q-Former in BLIP-2, or MLP in ClipCap) and the vision encoder (shown in Figure 1 *right*). We realize that the first stage, optimization of $p$ given $D_{\text{language}}$ and $t$, is purely linguistic and does not restrict the learning text examples from $\mathcal{D}_{\text{VL}}$. Thus, we propose to learn this part independently with the available sentence dataset.

While it's not feasible to learn individual prompts $p$ for each sentence $t$ due to the infinite number of possible sentences, we propose to parameterize prompt $p$ by a Prompting-Transformer (P-Former): $p = E_{\text{P-Former}}(t)$. This effectively transforms the learning of $p$ given $D_{\text{language}}$ and $t$ into learning $E_{\text{P-Former}}$ by $\arg\min_{E_{\text{P-Former}}} \mathcal{L}(D_{\text{language}}(E_{\text{P-Former}}(t)), t)$. Essentially, this is an autoencoder with the causal LLM $D_{\text{language}}$ as the decoder. As for P-Former, we use a bidirectional Transformer and the [CLS] representation as the bottleneck. Besides the reconstruction loss, we add a contrastive loss to discriminate each sample. Such a design makes $E_{\text{P-Former}}$ a semantic sentence embedding model like SimCSE [16] (i.e., semantically similar sentences have similar representations). Once $E_{\text{P-Former}}$ is learned, $p = E_{\text{P-Former}}(t)$ will be the "reference prompt" for LLM $D_{\text{language}}$ to generate $t$ auto-regressively. The training overview and P-Former details are shown in Figure 2.

Returning to the VL pre-training, we add a complementary loss to minimize the distance between aligned visual features (being used as language prompts) and the "reference prompt" given by P-Former. We expect this to improve the VL pre-training in two ways: (1) We further decouple the VL learning into another stage, as Li et al. [34] suggest that multi-stage training is important to mitigate alignment challenges. (2) A semantically rich space is learned for aligned visual features/prompts by a SimCSE design for our P-Former trained with the unimodal sentence dataset (i.e., semantically similar images are encouraged to align to "reference prompts" with close representations).

Our proposed framework only adds a learning objective on tensors feeding into LLMs as prompts (a.k.a images/multi-modalities as foreign languages [6, 61]). Therefore, our method is agnostic to the input modalities, X encoders, and X-to-language modules (where X can be images, videos, and audio). This could be especially salient for videos, which have much less high-quality paired data [15] compared to image-text pairs. And because P-Former is only trained with the LLM, there is no need to re-train the P-Former for different modalities.

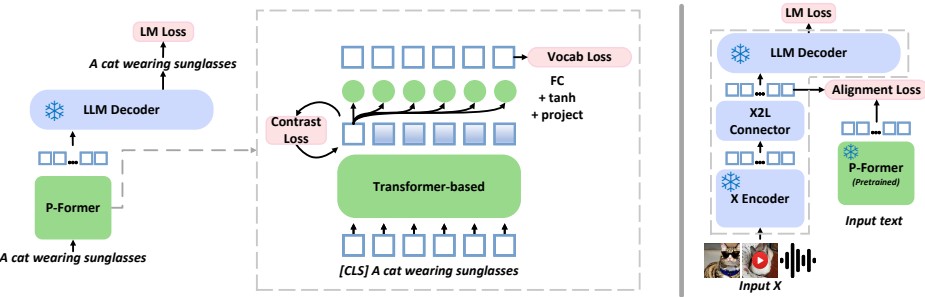

Figure 2: Overview of P-Former. **left:** The P-Former training resembles an autoencoder, with the bidirectional P-Former as the encoder and a causal LLM (frozen) as the decoder. The objective is to reconstruct input text auto-regressively. The [CLS] representation serves as sentence embeddings, which are projected back to the length of prompts. The contrastive loss at [CLS] mirrors the training of SimCSE [16]. A regularization vocabulary loss is utilized to encourage the prompts to be close to the vocabulary embeddings. **right:** Overview of bootstrapping VL pre-training with the trained P-Former. The alignment loss introduced by P-Former is agnostic to input modalities, encoders, and X-to-language modules (i.e., modules within the dashed box can be flexible). P-Former is only used during training and not during inference.

In our experiments, we take BLIP-2 as an example and show that our proposed framework improves this latest VL method by great margins in various benchmarks of VQA and image captioning. In Section 4.5, we demonstrate its effectiveness in other modalities (i.e., video) using different vision-to-language modules (i.e., plain Transformer over Q-Former).

We anticipate a growing body of future work within the paradigm of "images/multi-modalities as language prompts with frozen LLMs" due to its simplicity and effectiveness, as demonstrated by BLIP-2. For example, a concurrent work X-LLM [6] extends BLIP-2 from images to videos/speech with more advanced LLMs, augmenting BLIP-2's vision-to-language module Q-Former with Adapters. Because our proposed method is agnostic to input modalities, encoders, and X-to-language modules, it should seamlessly apply to future work within this paradigm of "images/multi-modalities as language prompts with frozen LLMs".

## 2   Related work

**End-to-end vision-language learning**   Most end-to-end VL pre-training models can be broadly classified into two categories: dual-encoder and fusion-encoder models. Dual-encoder models employ two separate networks for vision and language, with the modality interaction computed via dot-product between visual and linguistic features (e.g., CLIP [50]). Due to the efficient computation of vector dot-product through feature caching, dual-encoder models are effective and highly efficient for image-text retrieval tasks. However, their performance in VQA, captioning, and visual reasoning tasks is limited due to the lack of fine-grained alignment between the two modalities.

Fusion-encoder models, such as ALBEF [32], VLMo [4], and CoCa [69], introduce new fusion-Transformer layers to model deep interactions between the two modalities in addition to vision and language encoders. Common designs include concatenating visual and linguistic features before feeding them into a self-attentive Transformer [4, 7, 8, 14, 19, 20, 25, 27, 35, 37, 38, 54, 56, 59, 60, 61, 63, 66, 68, 71] or cross-attending vision and language encoders to compute fused features [2, 11, 12, 30, 32, 33, 40, 43, 44, 57, 65]. The vision encoder can range from simple linear embeddings [27] and ConvNets [19, 20, 25, 54, 60, 63, 68] to Transformers [4, 11, 12, 32, 33, 59, 61, 66], an offline pre-trained object detector like Faster-RCNN [7, 8, 14, 35, 37, 38, 56, 71], or an ensemble of models [42]. The language encoder can be initialized with a BERT-based [26] model or as part of a fusion-Transformer [4, 11, 12, 61, 70]. Most methods utilize three types of losses during pre-training: image-text contrastive (ITC) loss, image-text matching (ITM) loss, and mask language modeling (MLM) loss or language generation (ITG) loss. Fusion-encoder models have shown superior performance in VQA and captioning tasks, though they are less efficient in retrieval tasks. A thorough review of the recent advancements in VL pre-training can be found in Gan et al. [15].

**Vision-language learning with frozen language models**   Large language models, pre-trained on large text corpora, show exceptional performance in language generation tasks. Therefore, incorporating these large frozen language models into VL models can be particularly beneficial for vision-language generation tasks, such as VQA and captioning. Flamingo [2] incorporates visual signals into each layer of a large frozen LLM using cross-attention. In contrast, Frozen [58] fine-tunes the image encoder to align visual features as soft prompts, which are input into the frozen language model. Recently, BLIP-2 [34] introduced an additional vision-to-language adaptation module Q-former (in conjunction with the frozen ViT [10] and an LLM), proposing a two-stage training process to mitigate the challenges in learning visual-language alignment. The first stage of BLIP-2 training optimizes the Q-former to extract beneficial visual features using ITC, ITM, and ITG losses. In the second stage of BLIP-2 training, all three modules (ViT, Q-former, and LLM) are trained end-to-end with only the parameters in Q-former updated. Despite being trained on 129M image-text pairs and with affordable computational resources, BLIP-2 demonstrates competitive results across multiple benchmarks. Finally, a concurrent work on visual chat-bot X-LLM [6] also adopts a similar architectural design philosophy to BLIP-2. *Our proposed framework with P-Former can be applied to models under this paradigm that use soft prompts as the visual-language interface (e.g., Frozen, BLIP-2, X-LLM, etc).*

**Multi-modal auxiliary data learning**   Besides using off-the-shelf pre-trained vision encoders (ViT and Faster-RCNN [17, 51]) and language models, it is also interesting to explore how unimodal training can enhance multi-modal models. VLMo [4] demonstrated the benefits of conducting stage-wise pre-training with image-only and text-only data for their proposed model architecture. Li et al. [36] proposed using object tags from detectors as anchor points to bridge unpaired images and text, while Zhou et al. [74] formed pseudo-image-text pairs using an image-text retrieval alignment. Video-language models also leverage image-text pairs by repeating images to create static videos, constructing auxiliary paired datasets for pre-training. Jian et al. [22] showed that contrastive visual learning could also enhance contrastive sentence embeddings, a purely linguistic task. *We also show how pure language training can enhance a multi-modal model.*

## 3   Methodology

**Problem formulation**   Given an image-text dataset $\{I, t\} \in \mathcal{D}_{\text{VL}}$ and a unimodal language dataset composed purely of sentences $\{t\} \in \mathcal{D}_{\text{L}}$, our objective is to optimize the pre-training of a vision-language (VL) model. This model consists of a pre-trained vision encoder $E_{\text{vision}}$, a vision-to-language adaptation module $\underset{\text{V}\to\text{L}}{\Theta}$, and a frozen pre-trained language decoder $D_{\text{language}}$. The goal is to minimize the image-conditioned language generation loss, given that the vision encoder $E_{\text{vision}}$ is also frozen:

$$\underset{\underset{\text{V}\to\text{L}}{\Theta}}{\arg\min} \, \mathcal{L}_{\text{CrossEntropy}}(D_{\text{language}}(\underset{\text{V}\to\text{L}}{\Theta}(E_{\text{vision}}(I))), t) \tag{1}$$

As Li et al. [34] have noted, end-to-end optimization of Equation 1, visualized in Figure 1 *left*, can sometimes lead to catastrophic forgetting in LLMs.

### 3.1   Backward-decoupling and soft prompt pre-training (Training P-Former)

Let's denote the adapted visual features as $p = \underset{\text{V}\to\text{L}}{\Theta}(E_{\text{vision}}(I))$, which serve as soft prompts for the LLM $D_{\text{language}}$. During the optimization, Equation 1 can be decomposed into two parts, visualized in Figure 1 *right*:

$$\underset{p}{\arg\min} \, \mathcal{L}_{\text{CrossEntropy}}(D_{\text{language}}(p), t) \tag{2}$$

$$\underset{\underset{\text{V}\to\text{L}}{\Theta}}{\arg\min} \, \mathcal{L}_{\text{MSE}}(\underset{\text{V}\to\text{L}}{\Theta}(E_{\text{vision}}(I)), p) \tag{3}$$

Equation 2 essentially asks *"What is the optimal soft prompt $p$ that enables the auto-regressive language model $D_{language}$ to generate the sentence $t$."* Like all gradient-based deep learning models, depending on the training dataset, learning $p$ given $\{D_{\text{language}}, t\}$ could lead to different sub-optimal points[1] (a conventional deep learning problem is usually learning $D_{\text{language}}$ given $\{p, t\}$). End-to-end

---

[1]It can be easily verified that there exist multiple different soft prompts for an LLM to generate the same text auto-regressively. In an extreme example, a prompt with 32 tokens and a prompt with 16 tokens padded with 16 empty tokens (zeros vectors) can be both optimized for a LLM to generate the same text.

learning of Equation 1 can only use text $t$ from image-text dataset $\mathcal{D}_{\text{VL}}$ to update its intermediate variable $p$. However, we observe that the learning of Equation 2 involves no image, thus allowing us to leverage abundantly available unimodal sentences in $\mathcal{D}_{\text{L}}$.

Learning $p$ for each $t$ in $\mathcal{D}_{\text{L}}$ without constraint is intractable. Thus, we model $p$ by a bidirectional Transformer $E_{\text{P-Former}}$ (named Prompt-Former, or P-Former) $p = E_{\text{P-Former}}(t)$. Specifically, we use the output [CLS] hidden state of BERT as a compact representation for $t$ and project it back to the token length of $p$. Equation 2 can thus be reformulated as:

$$\underset{E_{\text{P-Former}}}{\arg\min} \mathcal{L}_{\text{CrossEntropy}}(D_{\text{language}}(E_{\text{P-Former}}(t)), t) \tag{4}$$

In essence, Equation 4 describes the training of an autoencoder with the bidirectional P-Former $E_{\text{P-Former}}$ serving as the encoder, and the auto-regressive LLM $D_{\text{language}}$ as the decoder. To enhance our model, we include an unsupervised contrastive loss $\mathcal{L}_{\text{contrast}}$, acting on the [CLS] representations of sentences to differentiate distinct instances. This loss, combined with our P-Former design, emulates the training of SimCSE [16], a semantic sentence embedding model (i.e., for semantically similar image-text pairs, the predicted prompts by P-Former should also be close). Furthermore, we introduce a regularization loss $\mathcal{L}_{\text{vocab}}$ to minimize the distance between each token in $p$ and the closest embedding of the LLM's ($D_{\text{language}}$) vocabularies. The final objective becomes:

$$\underset{E_{\text{P-Former}}}{\arg\min}(\mathcal{L}_{\text{CrossEntropy}}(D_{\text{language}}(E_{\text{P-Former}}(t)), t) + \mathcal{L}_{\text{contrast}} + \mathcal{L}_{\text{vocab}}) \tag{5}$$

A comprehensive view of the P-Former's architecture and learning losses is presented in Figure 2 *left*. We emphasize that the optimization of Equation 5 and P-Former training rely only on the text. Upon training the P-Former, Equation 3 can be reformulated as:

$$\underset{\underset{\text{V}\rightarrow\text{L}}{\Theta}}{\arg\min} \mathcal{L}_{\text{MSE}}(\underset{\text{V}\rightarrow\text{L}}{\Theta}(E_{\text{vision}}(I)), E_{\text{P-Former}}(t)) \equiv \underset{\underset{\text{V}\rightarrow\text{L}}{\Theta}}{\arg\min} \mathcal{L}_{\text{alignment}} \tag{6}$$

This new form, depicted in Fig 2 *right*, minimizes the distance between the aligned visual features and the prompts predicted by the trained P-Former, effectively aligning visual-linguistic representations.

## 3.2 Preliminary: BLIP-2 forward-decoupled training

While our proposed framework is flexible in regards to the specific architecture of $\underset{\text{V}\rightarrow\text{L}}{\Theta}$ or the learning strategy deployed, for illustrative purposes, we employ BLIP-2 as a case study to demonstrate the applicability of our approach with state-of-the-art learning methods, owing to the strong performance and reproducibility of BLIP-2. In the context of BLIP-2, $E_{\text{vision}}$ is a ViT-g, $\underset{\text{V}\rightarrow\text{L}}{\Theta}$ is referred to as Q-Former, and $D_{\text{language}}$ is a OPT$_{2.7\text{B}}$. BLIP-2 proposes a two-stage pre-training process, with the initial stage involving the pre-training of $\underset{\text{V}\rightarrow\text{L}}{\Theta}$ by:

$$\underset{\underset{\text{V}\rightarrow\text{L}}{\Theta}}{\arg\min} \text{ITC}(\underset{\text{V}\rightarrow\text{L}}{\Theta}(E_{\text{vision}}(I)), \underset{\text{V}\rightarrow\text{L}}{\Theta}(t)) + \text{ITM}(\underset{\text{V}\rightarrow\text{L}}{\Theta}(E_{\text{vision}}(I), t)) + \text{ITG}(\underset{\text{V}\rightarrow\text{L}}{\Theta}(E_{\text{vision}}(I), t)) \tag{7}$$

This is followed by a second stage that involves end-to-end training of Equation 1. The terms ITC, ITM, and ITG in Equation 7 are utilized to guide the Q-Former $\underset{\text{V}\rightarrow\text{L}}{\Theta}$ in extracting visually relevant features that correspond to the associated captions. We refer to this two-step process in BLIP-2 – first determining the visual features to extract and then incorporating the selected visual features into an end-to-end learning framework – as "forward-decoupled training."

## 3.3 BLIP-2 forward-decoupled training with pre-trained P-Former

We now describe the full training pipeline when integrating our framework with BLIP-2. The first stage of training involves pre-training the Q-Former with Equation 7 ($\mathcal{L}_{\text{BLIP2-stage1}} \equiv \text{ITC} + \text{ITM} + \text{ITG}$), supplemented with the alignment loss introduced by the P-Former, as defined in Equation 6:

$$\mathcal{L}_{\text{BLIP2-stage1}} + \omega_1 \times \mathcal{L}_{\text{alignment}} \tag{8}$$

Subsequently, the second stage of training, in line with our approach, involves BLIP-2's stage 2, which is the end-to-end training of Equation 1: $\mathcal{L}_{\text{BLIP2-stage2}} \equiv \mathcal{L}(D_{\text{language}}(\underset{\text{V}\rightarrow\text{L}}{\Theta}(E_{\text{vision}}(I))), t)$, again enhanced with the alignment loss imparted by P-Former in Equation 6:

$$\mathcal{L}_{\text{BLIP2-stage2}} + \omega_2 \times \mathcal{L}_{\text{alignment}} \tag{9}$$

Figure 3 provides a schematic representation of the proposed integration of our framework and P-Former with BLIP-2.

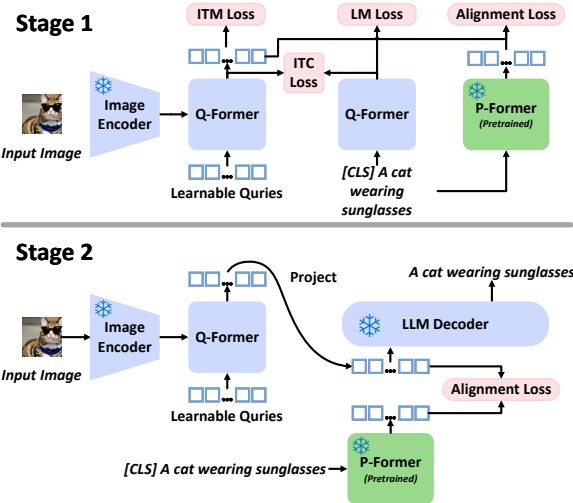

Figure 3: An overview of our framework with BLIP-2, which employs a two-stage training process. The green components represent the alignment loss and modules added by us, which do not require gradients. The blue components are part of the original BLIP-2 structure. **P-Former is solely utilized during training and is not required during the inference phase.** Our proposed framework, with P-Former, can be seamlessly applied to any models that leverage prompts as the interface for multi-modal-language communications.

## 3.4 Model pre-training

**Training dataset** We employ a 12M subset of the pseudo-labeled [33] LAION dataset [52], using only the sentences, for pre-training the P-Former. For VL pre-training, we widely adapted academic setting (since academic institutions lack the resources available to industry researchers to use very large datasets) with approximately 4M image-text pairs. This set comprises the MSCOCO-80K [39], VG-100K [28], CC-3M [53], and SBU-1M [47] datasets.

**Pre-training models** Our method is universally applicable to any vision-to-text models that utilize prompts as the interface. Owing to its impressive performance and reproducibility, we chose BLIP-2 as the base model for our primary experiments. Thus, for VL pre-training, the image encoder $E_{\text{vision}}$ is a ViT-g/14 from EVA-CLIP [13], the LLM decoder $D_{\text{language}}$ is an OPT$_{2.7B}$ [72], and the vision-to-language adaptation module is a Q-Former [34]. The Q-Former is initialized by BERT-base with 32 learnable queries. Our newly proposed P-Former is a base Transformer initialized by BERT-base.

**Pre-training details** The P-Former is trained on a system with $3 \times$ RTX-A6000 (48GB) GPUs, using PyTorch [48]. We trained for five epochs with a linear warm-up and cosine scheduling, using a batch size of 384 ($3 \times 128$), and AdamW as the optimizer. The initial learning rate is set to $1e^{-4}$, with a minimum learning rate of $1e^{-5}$, a warm-up learning rate of $1e^{-6}$, and 2000 warm-up steps. The VL pre-training is performed on a server equipped with $8 \times$ RTX-A6000 (48GB) GPUs, using PyTorch. We developed the code based on the LAVIS project [31]. Predominantly, we employed the default configuration files provided by BLIP-2 of LAVIS. Both the stage 1 and stage 2 training ran for 10 epochs with linear warm-up and cosine scheduling, using a batch size of 1024 ($8 \times 128$), and AdamW as the optimizer. The weight decay is set to 0.05, the initial learning rate is $1e^{-4}$, the minimum learning rate is $1e^{-5}$, and the warm-up learning rate is $1e^{-6}$. The key distinction is that stage 1 and stage 2 incorporate 5000 and 2000 warm-up steps, respectively. We set $\omega_1 = 10$ and $\omega_2 = 100$ while training BLIP-2 OPT$_{2.7B}$ with our P-Former.

**Computational overhead considerations** Incorporating $\mathcal{L}_{\text{alignment}}$ from Equation 8 and 9 introduces only a minimal computational overhead, attributable to an additional forward pass of the P-Former (Transformer-base) at each iteration. To illustrate, in our experimental settings using BLIP-2 OPT$_{2.7B}$, the training time for stage 1 saw a modest increase from 2,669 minutes to 2,743 minutes. Similarly, for stage 2, the training time increased marginally from 1,862 minutes to 1,880 minutes. Thus, our methodology's overall computational burden remains manageable despite its enhancements (the only additional cost is pre-training of the P-Former, which only needs to be done once for an LLM).

# 4 Experiments

Given the impressive performance and accessibility of the BLIP-2 model, coupled with its open-source nature, we primarily employ it as our base model. We aim to demonstrate how our proposed "backward-decoupling" strategy, along with the learned P-Former, can enhance the baselines across various image-to-text generation benchmarks. In Section 4.5, we further extend the applicability of our framework to other modalities, utilizing different base models.

## 4.1 Zero-shot image-to-text generation

We assess the performance of our pre-trained models on zero-shot VQA, encompassing GQA [21], OKVQA [45], and VQAv2 [18], without any task-specific fine-tuning. As per BLIP-2, we append text prompts to visual prompts prior to their processing by the frozen LLM. Both for the baseline BLIP-2 and our model, the text prompt used is "Question:  Short answer:". The results, as detailed in Table 1, suggest that our proposed framework significantly enhances the zero-shot VQA performance of BLIP-2 trained with 4M image-text pairs. Remarkably, the gap between the BLIP-2 trained with 4M and 129M image-text pairs is largely bridged by our method.

| Models | #Pretrain Image-Text | Pretrain Uni-Text | VQAv2 val | VQAv2 test-dev | OK-VQA test | GQA test-dev |
|---|---|---|---|---|---|---|
| FewVLM [24] | 9.2M | - | 47.7 | - | 16.5 | 29.3 |
| Frozen [58] | 3M | - | 29.6 | - | 5.9 | - |
| VLKD [9] | 3M | - | 42.6 | 44.5 | 13.3 | - |
| Flamingo3B [2] | 1.8B | - | - | 49.2 | 41.2 | - |
| OPT$_{2.7B}$ BLIP-2 [34] | 4M | - | 46.8 | 45.6 | 25.9 | 30.5 |
| OPT$_{2.7B}$ Ours | 4M | ✓ | _52.6_ | _52.2_ | _30.0_ | _34.0_ |
| OPT$_{2.7B}$ BLIP-2$^{\dagger}$ [34] | 129M | - | **53.5** | **52.3** | **31.7** | **34.6** |

Table 1: Comparison with different methods on zero-shot VQA $^{\dagger}$: numbers taken from Li et al. [34].

## 4.2 Fine-tuned image captioning

We further fine-tune our pre-trained model for MSCOCO [39] image captioning, employing the text prompt "a photo of ". Following BLIP-2, we fine-tune the model for 5 epochs using a batch size of 1024 ($8 \times 128$), AdamW with an initial learning rate of $1e^{-5}$, minimum learning rate of 0, warm-up learning rate of $1e^{-8}$ and 1000 warm-up steps, with linear warm-up and cosine scheduling. We evaluate our fine-tuned model on the Karpathy test split of MSCOCO. Also, zero-shot transfer results on the NoCaps dataset [1] are reported. Shown in Table 2, our framework improves BLIP-2 in all metrics, with greater improvements in CIDEr compared to SPICE.

| Models | #Pretrain Image-Text | NoCaps Zero-shot (validation set) in-domain C | S | near-domain C | S | out-domain C | S | overall C | S | COCO Fine-tuned Karpathy test B@4 | C |
|---|---|---|---|---|---|---|---|---|---|---|---|
| OSCAR [38] | 4M | - | - | - | - | - | - | 80.9 | 11.3 | 37.4 | 127.8 |
| VinVL [71] | 5.7M | 103.1 | 14.2 | 96.1 | 13.8 | 88.3 | 12.1 | 95.5 | 13.5 | 38.2 | 129.3 |
| BLIP [33] | 129M | 114.9 | 15.2 | 112.1 | 14.9 | 115.3 | 14.4 | 113.2 | 14.8 | 40.4 | 136.7 |
| OFA [60] | 20M | - | - | - | - | - | - | - | - | 43.9 | 145.3 |
| Flamingo [2] | 1.8B | - | - | - | - | - | - | - | - | - | 138.1 |
| SimVLM [63] | 1.8B | 113.7 | - | 110.9 | - | 115.2 | - | 112.2 | - | 40.6 | 143.3 |
| OPT$_{2.7B}$ BLIP-2 [34] | 4M | 115.3 | 15.0 | 111.0 | 14.6 | 112.5 | 14.0 | 111.9 | 14.5 | 41.8 | 140.4 |
| OPT$_{2.7B}$ Ours | 4M | _118.3_ | _15.3_ | _114.7_ | _14.9_ | _114.1_ | _14.1_ | _115.1_ | _14.8_ | _42.3_ | _141.8_ |
| OPT$_{2.7B}$ BLIP-2$^{\dagger}$ [34] | 129M | **123.0** | **15.8** | **117.8** | **15.4** | **123.4** | **15.1** | **119.7** | **15.4** | **43.7** | **145.8** |

Table 2: Comparison with different captioning methods on NoCaps and COCO. All methods optimize the cross-entropy loss during fine-tuning. C: CIDEr, S: SPICE, B: BLEU. $^{\dagger}$: numbers taken from Li et al. [34].

## 4.3 Zero-shot image-text retrieval

While our proposed method primarily focuses on refining visual prompts for a frozen LLM to generate corresponding text, it may not prove as beneficial for image-text retrieval tasks (the ITC and ITM losses are principally responsible for these tasks). Nevertheless, we present results on zero-shot

MSCOCO, and zero-shot Flickr30K [49] image-to-text and text-to-image retrievals. We compare two models trained with $\mathcal{L}_{\text{BLIP2-stage1}}$ (ITC, ITM and ITG) and $\mathcal{L}_{\text{BLIP2-stage1}} + \mathcal{L}_{\text{alignment}}$, without any further task-specific fine-tuning. As expected, Table 3 reveals that the newly introduced $\mathcal{L}_{\text{alignment}}$ offers limited benefits for retrieval tasks. However, it does not negatively impact the performance.

| Task | Pre-training objectives | Image → Text R@1 | R@5 | Text → Image R@1 | R@5 |
|---|---|---|---|---|---|
| Flickr30K | $\mathcal{L}_{\text{BLIP2-stage1}}$ | **94.3** | **99.8** | 82.9 | 95.5 |
| | $\mathcal{L}_{\text{BLIP2-stage1}} + \mathcal{L}_{\text{alignment}}$ | 93.7 | 99.7 | **83.0** | **95.8** |
| MSCOCO | $\mathcal{L}_{\text{BLIP2-stage1}}$ | 78.4 | 93.8 | **60.5** | **83.0** |
| | $\mathcal{L}_{\text{BLIP2-stage1}} + \mathcal{L}_{\text{alignment}}$ | **78.7** | **94.5** | 60.4 | 82.8 |

Table 3: Comparison with different image-to-text and text-to-image retrieval methods.

## 4.4 Ablation studies

**Impact of alignment loss weights** We investigate the influence of $\omega_1$ and $\omega_2$ in Equation 8 and 9. $\omega_1 = 0$ and $\omega_2 = 0$ refers to BLIP-2, and $\omega_1 = 10$ and $\omega_2 = 100$ refers to our default configuration of BLIP-2 + P-Former. The alignment loss introduced by the P-Former proves beneficial in both stages of VL pre-training, as shown in Table 4.

**Alternate language model** In this section, we substitute the decoder-based $\text{OPT}_{2.7B}$ model with an encoder-decoder-based FLAN-T5$_{XL}$ as the new LLM. The experiments are conducted with a limited computational budget on $3 \times$ RTX-A6000 and for 5 epochs on both stage 1 and stage 2. The results, displayed in Table 5, verify the effectiveness of our framework with another LLM.

| $\omega_1$ | $\omega_2$ | VQAv2 val | OK-VQA test | GQA test-dev |
|---|---|---|---|---|
| 0 | 0 | 46.8 | 25.9 | 30.5 |
| 10 | 0 | 51.4 | 29.2 | 32.8 |
| 0 | 100 | 50.4 | 28.7 | 33.0 |
| 10 | 100 | **52.6** | **30.0** | **34.0** |

Table 4: Ablations on $\omega_1$ and $\omega_2$ of Equation 8 and 9 (using $\text{OPT}_{2.7B}$ as LLMs).

| Models | #Pretrain Image-Text | VQAv2 val | OK-VQA test | GQA test-dev |
|---|---|---|---|---|
| Flan-T5$_{XL}$ BLIP-2$^‡$ | 4M | 48.3 | 31.5 | 36.4 |
| Flan-T5$_{XL}$ ours$^‡$ | 4M | 54.9 | 35.7 | 40.3 |
| Flan-T5$_{XL}$ BLIP-2$^†$ | 129M | **62.6** | **39.4** | **44.4** |

Table 5: Experiments using Flan-T5$_{XL}$ as LLM. $^‡$: using much less GPUs/epochs compared to Sec.4.1. $^†$: from Li et al. [34].

**Effect of P-Former's pre-training sentence datasets** In our primary experiments, we utilize a dataset containing 12M sentences for P-Former training. We investigate the impact of the pre-training sentence dataset for P-Former by re-training it with 4M sentences from our VL pre-training datasets. We then train BLIP-2 + P-Former and report zero-shot VQA results in Table 6. This examination underscores that both the implicit decoupling of BLIP-2's two-stage training into a 3-stage training (pre-training of P-Former), and the employment of additional unimodal sentences contribute to the improved outcomes.

| P-Former | #Pretrain Sentences | VQAv2 val | OK-VQA test | GQA test-dev |
|---|---|---|---|---|
| × | - | 46.8 | 25.9 | 30.5 |
| ✓ | 4M | 51.7 | 28.2 | 32.3 |
| ✓ | 12M | **52.6** | **30.0** | **34.0** |

Table 6: Ablations on sentence datasets used to train P-Former (using $\text{OPT}_{2.7B}$ as LLMs). The first row w/o P-Former is baseline BLIP-2.

| | BLEU-4 | CIDEr | ROUGE |
|---|---|---|---|
| NITS-VC [55] | 20.0 | 24.0 | 42.0 |
| ORG-TRL [73] | 32.1 | 49.7 | 48.9 |
| $\mathcal{L}_{\text{ITG}}$ | 29.3 | 56.6 | 48.2 |
| $\mathcal{L}_{\text{ITG}} + \mathcal{L}_{\text{alignment}}$ | **30.9** | **60.9** | **49.1** |

Table 7: VATEX English video captioning. Baseline is a sequential model (I3D → Transformer → $\text{OPT}_{2.7B}$), training end-to-end with ITG.

## 4.5 Video captioning

Our framework is modality-agnostic with respect to the visual encoder and vision-to-language adaptor, making it applicable to other modalities, such as video. Consequently, we establish a video learning

pipeline, with the vision encoder set as a frozen I3D [5] video encoder, the vision-to-language adaptor as a Transformer-base, and the LLM decoder as the $OPT_{2.7B}$ (also frozen). We then train this model on the VATEX [62] English training set and evaluate it on the validation set. This dataset contains 26K videos for training. The experiments are conducted on an RTX-A6000. Initially, we train the model solely using $\mathcal{L}_{\text{alignment}}$ for 10 epochs with the P-Former, followed by end-to-end learning with $\mathcal{L}_{\text{ITG}}$ for an additional 10 epochs.

Our baseline, represented in Table 7, is competitive with two well-established video captioning models: MITS-VC [55] and ORG-TRL [73]. It is noteworthy that the current state-of-the-art on this benchmark, VideoCoCa [67], is trained on 10M videos, in contrast to our model, which is trained on merely 26K videos. Furthermore, the integration of P-Former and $\mathcal{L}_{\text{alignment}}$ enhances the CIDEr score by $4.3$ (from $56.6 \rightarrow 60.9$).

Despite being a smaller-scale experiment without large-scale pre-training, we demonstrate that our learning framework can be generalized to another modality (i.e., video-learning), employing a different vision-language adaptor (i.e., a plain Transformer as opposed to a Q-Former).

## 5   Limitations

Despite the modality-agnostic nature of P-Former and its ability to adapt to various encoders and vision-to-language adaptors, the unimodal language pre-training remains contingent on the choice of the frozen LLM. This necessitates re-training of the P-Former for different language decoders such as $OPT_{2.7B}$ and $FLAN-T5_{XL}$. Moreover, incorporating P-Former primarily enhances image-to-text generation tasks such as VQA and image captioning, while it falls short in improving image-text retrieval tasks. Finally, our methodology primarily assists in bootstrapping prompt-based VL pre-training, i.e., providing aligned visual features as soft prompts to LLMs. Its application to Flamingo remains unclear due to its cross-attention basis and non-open-source status. Nevertheless, given the simplicity of sequential modules of prompt-based models (as demonstrated by recent works such as Frozen, BLIP-2, X-LLM, etc.), we anticipate that our framework will be broadly applicable to most future work in the academic setting.

## 6   Conclusion and discussion

This paper introduces a novel optimization framework for enhancing vision-language models based on large, frozen LLMs. We observe that the end-to-end image-to-text pre-training can be backwardly decoupled: initially determining the "ideal prompt" that triggers the LLM to generate the target text (which can be trained in an unsupervised fashion), followed by the alignment of visual features to the prompt. To this end, we train a P-Former, which functions similarly to a semantic sentence embedding model, to predict prompts to which visual features should align. Experimental results demonstrate that including alignment loss (via P-Former) in the BLIP-2's framework significantly narrows the performance gap between models trained with 4M and 129M image-text pairs.

The key contributions of this paper are as follows:

- Contrary to most prior studies, which decouple VL pre-training into (1) learning which visual features to forward into language modules and (2) conducting end-to-end learning with the selected visual features (dubbed "forward-decoupling"), we propose an innovative perspective of VL decoupled-training from a backward viewpoint. We bifurcate the training into (1) determining the "ideal prompt" for the LLM to generate the text and (2) aligning visual features to that prompt.
- We introduce the P-Former, designed to predict the "ideal prompt," which is trained using a unimodal sentence dataset. This exhibits a novel application of unimodal training in enhancing multi-modal learning.
- Our proposed training framework substantially enhances a robust and recent baseline (BLIP-2), bridging the gap between models trained with 4M and 129M image-text pairs using accessible hardware ($8 \times$ RTX-A6000 in less than 4 days). This considerably lowers the entry barriers to VL pre-training research and is expected to attract interest from groups with limited resources.
- The proposed framework generally applies to different modalities (images, videos, audio, etc.), vision encoders, and vision-to-language modules.

Lastly, we address the commonly asked questions by the reviewers in Appendix A, B, C, D, and E.

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

## A    Intuition and motivation behind P-Former

In this section, we summarize the intuitive explanation and motivation on why learning an ideal language prompt helps more than using visual ones as in the counterpart models.

- In our experiments with base models like BLIP-2, the architecture consists of three sequential components: (1) ViT, (2) VL-connector, and (3) LLM decoder. Since we use a frozen LLM for generation, optimizing closer to the LLM decoder becomes more pivotal for achieving optimal generation quality.
- The unique design of P-Former mirrors a sentence embedding model. This means the prompts predicted by the P-Former carry rich semantics. Therefore, during evaluations on unfamiliar images, the model boasts an improved generalization capability.
- BLIP2's studies indicate that direct end-to-end optimization of the sequential model can sometimes lead to catastrophic forgetting. Our approach adds an additional layer of complexity by decomposing the 2-stage BLIP2 training into 3 stages, further addressing this optimization challenge.
- For BLIP2, optimization of soft prompt is learned only using text from image-text pair, while our decoupled training allows for leveraging additional unimodal data for optimizing these soft prompts

## B    Justification for lack of ablation experiments w/ and w/o the P-Former

We purposely omitted experiments with and without the P-Former module (e.g., using a randomly initialized prompt $p$). This omission was driven by the following considerations:

- **Random initialization and learning without P-Former**: Our initial approach was to directly learn from a randomly initialized prompt $p$ without incorporating the P-Former. But, upon testing, we identified a significant challenge. For a smaller model variant like opt-2.7b, which possesses a hidden size of 2560, if we employ 32 tokens as soft prompts for an expansive dataset with 4M sentences, the resultant model would have to accommodate an overwhelming 327B parameters. This would have computational implications and potentially overfit, as learning from such a vast parameter space can dilute the essential semantic connections between various sentences.
- **P-Former's efficiency in parameterization**: The P-Former emerged as a solution to this parameter explosion problem. Instead of requiring a unique prompt for each data point in the dataset, the P-Former parameterizes the soft prompt p using a semantically-rich Transformer model. This design ensures that the total number of parameters remains fixed at 110M. The major advantage here is scalability. Whether working with a dataset of 4M, 12M, or even larger (e.g., 129M) or LMs with varying decoder sizes, the P-Former guarantees a consistent number of parameters, making the model more computationally efficient and preventing the loss of essential semantic relationships.

In brief, our experimentation strategy was driven by the dual goals of maintaining computational efficiency while preserving rich semantics. The challenges posed by direct learning from a randomly initialized prompt emphasized the need for a more structured approach, leading to the birth of the P-Former concept.

## C    Qualitative analysis on VQA

In this section, we incorporate qualitative comparisons for the GQA and OKVQA datasets, allowing us to offer more nuanced insights. In Figure C.1, we show several examples comparing our model's response with BLIP-2 and the ground truth (GT). From these examples, it can be observed that there is greater agreement with GT by our model.

It should be noted that the abstract semantic reasoning of our model can sometimes lead to artificially low scores for our model when looking for an exact match. For instance, asking "What occupation might he have?" with a picture of a person driving a forklift generates the answer "forklift operator" by our model, whereas the correct exact answer in the GT is stated as "forklift driver." Though these two answers are semantically identical, they will count as a wrong generation by our model.

## D    Additional discussion of the results

In this section, we provide more interpretation of the results. For instance, Table 1, in addition to underscoring the potency of our proposed framework in bolstering the zero-shot VQA performance,

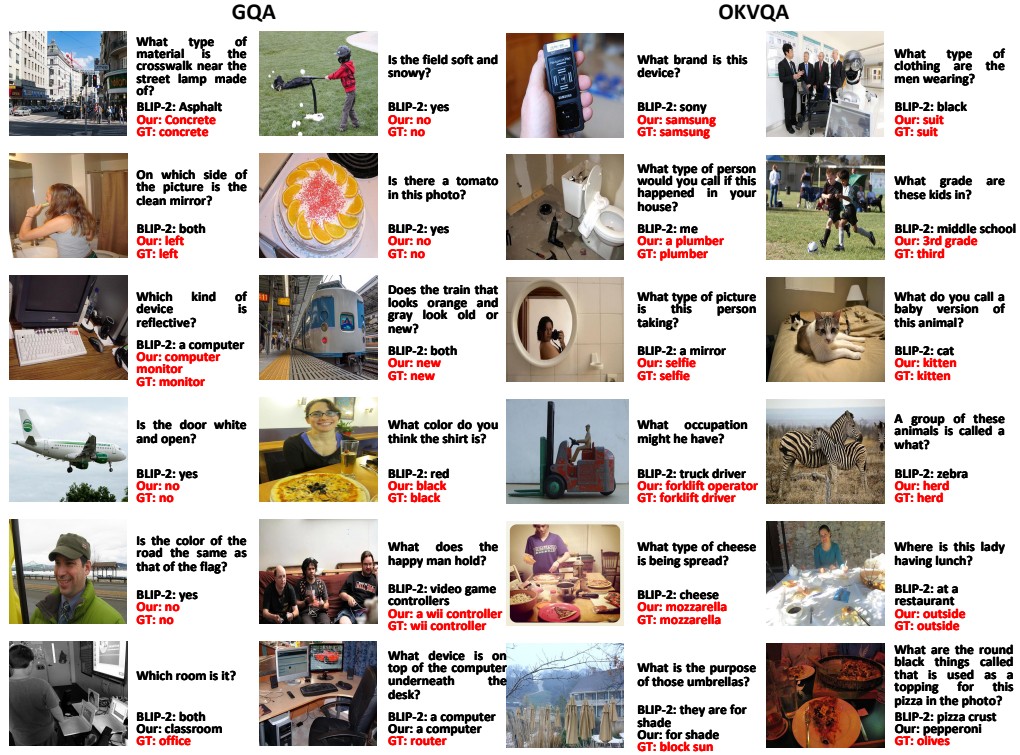

Figure C.1: Qualitative analysis on success and failure cases of GQA and OKVQA.

particularly when trained with 4M image-text pairs, shows that our method manages to considerably close the performance gap between the BLIP-2 trained on different scales: 4M and 129M image-text pairs. This suggests that the effectiveness of our model is not solely a function of the amount of training data but rather the methodology itself. In essence, this table illustrates how strategic modifications and improvements can achieve comparable results to models trained on much larger datasets.

Similarly, Table 2 provides insights into our model's adaptability. When we fine-tune our pre-trained model for a specific task like MSCOCO image captioning, the results reflect an overall enhancement over BLIP-2 across all metrics. The pronounced improvement in CIDEr, as opposed to SPICE, indicates that our model is adept at recognizing and generating more relevant and contextually accurate descriptions of images. The additional data on zero-shot transfer to the NoCaps dataset further substantiates the model's capability to generalize and adapt to newer, unseen data.

Finally, while our model's primary design goal is to refine visual prompts for text generation, Table 3 offers a perspective on its performance in the retrieval domain. Even though the model was not specifically optimized for retrieval tasks, it is evident that the introduced modifications do not compromise the retrieval performance, attesting to the model's robustness.

# E   LLM-dependence of the stage-1 pre-training

It should be noted that our stage-1 pre-training needs to be repeated for each LLM, if $\omega_1 \neq 0$. However, as evidenced in Table 4 ($\omega_1 = 0$ and $\omega_2 = 100$), our approach achieves competitive results even without the alignment loss in stage-1, focusing the alignment solely on stage-2.

# F   Acknowledgement

The authors would like to thank Mu Li and Yi Zhu from Boson AI for their YouTube and Bilibili videos on paper reading, which greatly inspired this work.

