# OpenReview forum: "Bootstrapping Vision-Language Learning with Decoupled Language Pre-training"
_NeurIPS.cc/2023/Conference — NeurIPS 2023 spotlight_

### Official Review · Reviewer_FD1R · 2023-06-14

**Soundness:** 3 good
**Presentation:** 2 fair
**Contribution:** 3 good
**Rating:** 6
**Confidence:** 5

**Summary:**

This paper propose to add a new component, a P-Former, to the BLIP-2 vision-language pre-training framework. The P-Former is a sentence encoder that learns to project texts into the input space of a LLM. During vision-language pre-training, an additional alignment loss is applied between BLIP-2 Q-Former's visual feature and the P-Former's text feature, which improves the alignment between the Q-Former and the frozen LLM, leading to improved image-to-text generation performance.

**Strengths:**

- This paper explores an interesting direction to improve BLIP-2 pre-training with an additional text encoder. The text encoder serves as a surrogate of the LLM to help better aligning the Q-Former with a frozen LLM. This could open up future research opportunities.

- The experiments show some good improvements w.r.t. the original BLIP-2 on image-to-text generation tasks, which verifies the improved alignment with the LLM. It is also good to see improvements with both OPT and FlanT5.

**Weaknesses:**

I enjoy the core idea of the paper. However, the paper still needs much improvement in multiple aspects as detailed below.

1. The motivation and problem formulation is a slight misrepresentation of the actual method. The actual role of the P-Former is to serve as a surrogate of the LLM to help the Q-Former better align with the frozen LLM in both stage-1 and stage-2 pre-training.

2. The proposed method has a significant conceptual difference from BLIP-2: the stage-1 pre-training is *not* LLM-agnostic. This means that the more expensive stage-1 pre-training needs to be performed for every LLM. The proposed method thus tradeoff the Q-Former's flexibility for better alignment with the LLM. This point needs to be highlighted.

3. The pre-training of the P-Former is rather complex, where the intuition of each training loss is not clear. There needs to be more discussion and ablation study on the multiple losses.

4. The effect of the P-Former alignment may or may not maintain given more pre-training data and longer pre-training epochs. This is hard to verify given limited compute, so I'm not expecting the authors to provide answers.

5. The writing could be improved. Personally, I feel many equations to be unnecessary because they add additional burdens to understand the method. Simple plain text would be good enough to describe the method clearly. The "forward-decoupled" and "backward-decoupled" concepts are also not intuitive to understand.

**Questions:**

- Is the P-Former finetuned during stage-1 and stage-2 pre-training?

**Limitations:**

The authors have addressed the limitations.

---

> ### Author Rebuttal · Authors · 2023-08-10
>
> **W1: The motivation and problem formulation is a slight misrepresentation of the actual method. The actual role of the P-Former is to serve as a surrogate of the LLM to help the Q-Former better align with the frozen LLM in both stage-1 and stage-2 pre-training.**
>
> Re: The P-Former does indeed function as a surrogate for the LLM, assisting the Q-Former to align optimally with the frozen LLM during both stage-1 and stage-2 pre-training. The driving force behind our methodology, however, was to develop a decoupled training approach. Our primary intention was to create an optimal prompt, learned by the P-Former, to guide the VL-connector (e.g., Q-Former) effectively.
>
> **W2: The proposed method has a significant conceptual difference from BLIP-2: the stage-1 pre-training is not LLM-agnostic. This means that the more expensive stage-1 pre-training needs to be performed for every LLM. The proposed method thus tradeoff the Q-Former's flexibility for better alignment with the LLM. This point needs to be highlighted.**
>
> Re: You've correctly identified a departure in our approach from BLIP-2: our stage-1 pre-training is indeed not LLM-agnostic. This difference will be highlighted  in our revised manuscript. However, as evidenced in the 3rd row of Table 4 (w1=0 and w2=100) from our ablation study, our approach achieves competitive results even without alignment loss in stage-1, focusing the alignment solely on stage-2.
>
> **W3: The pre-training of the P-Former is rather complex, where the intuition of each training loss is not clear. There needs to be more discussion and ablation study on the multiple losses.**
>
> Re: The contrastive loss is employed to enhance the semantic richness of the P-Former’s output, ensuring similar images/sentences yield similar prompts. Furthermore, our findings indicate that the inclusion of vocabulary loss (which can be viewed as a regularization term) slightly improves our results (without vocabulary loss: GQA:33.5, OKVQA: 29.5, VQA: 51.7. With vocabulary loss: GQA: 34.0, OKVQA: 30.0, VQA: 52.6).
>
> **W4: The effect of the P-Former alignment may or may not be maintained given more pre-training data and longer pre-training epochs. This is hard to verify given limited computation, so I'm not expecting the authors to provide answers.**
>
> Re: Conducting experiments with the 129M dataset presents challenges for us, given that we possess a max of 8 GPUs, while the original results reported by BLIP2 on the 129M dataset utilized 16 GPUs about 9 days. The primary goal of our method is to streamline the training process and make efficient use of the available training data. As such, we anticipate that our approach might show modest gains in a 129M setting, particularly if the model undergoes extensive training. In fact, a key motivation behind P-Former is to reduce the dependence on vast multi-modal datasets and models. This approach not only simplifies the training process but also democratizes participation, ensuring that research in this area isn't solely the domain of entities with access to significant computational resources.
>
> **W5: The writing could be improved. Personally, I feel many equations to be unnecessary because they add additional burdens to understanding the method. Simple plain text would be good enough to describe the method clearly. The "forward-decoupled" and "backward-decoupled" concepts are also not intuitive to understand.**
>
> Re: We are grateful for your feedback. We will revisit our manuscript to ensure greater clarity, potentially simplifying certain sections for enhanced comprehension. Complex concepts like “forward-decoupled” and “backward-decoupled” will be elaborated upon to provide a more intuitive understanding.
>
> **Q1: Is the P-Former fine-tuned during stage-1 and stage-2 pre-training?**
>
> A: The P-Former remains frozen during both stage-1 and stage-2. Learned during stage 0, the P-Former is designed to predict semantically rich soft prompts for individual instances during the subsequent stages. Its fixed state during these stages is pivotal for generating alignment losses, directing the outputs of the VL-connector (e.g., Q-Former in BLIP2).

---

> > ### Comment · Reviewer_FD1R · 2023-08-13
> >
> > I appreciate the authors' response. I confirm my original score and recommend acceptance.

---

> > > ### Author Response · Authors · 2023-08-14
> > > **Thank you!**
> > >
> > > Thank you! We appreciate the time and effort.

---

### Official Review · Reviewer_T7A1 · 2023-07-05

**Soundness:** 3 good
**Presentation:** 3 good
**Contribution:** 4 excellent
**Rating:** 7
**Confidence:** 4

**Summary:**

This paper introduces a vision-language pre-training method with the help of the proposed Prompt-Transformer. As the first step, the P-Former is optimized to learn the "optimal" soft prompt that can guide the LLM to generate the target texts. After that, the trained P-Former is frozen and used to train the visual adaptor of VLM.

**Strengths:**

1. The method is novel in that it utilizes an intermediate P-Former to effectively optimize the VLM.
2. The method achieves impressive results in terms of the performance of training data efficiency.

**Weaknesses:**

1. In Table 1, it is not fair to only list the numbers of image-text pairs considering that P-Former also consumes language data for training. Similarly, in the paragraph of Line 216, the overhead of training P-Former should also be taken into consideration.
2. After reading the paper, it is still a bit unclear why, in essence, P-Former is effective. Does it function as some type of knowledge distillation? Please provide more discussion about this.

**Questions:**

None

**Limitations:**

The authors discussed three limitations in the paper. I am mainly concerned about the last one that the method cannot handle cross-attention models. With the development of VLM, cross-attention or joint modeling could be a very important branch of methods.

---

> ### Author Rebuttal · Authors · 2023-08-10
>
> **W1: In Table 1, it is not fair to only list the numbers of image-text pairs considering that P-Former also consumes language data for training. Similarly, in the paragraph of Line 216, the overhead of training P-Former should also be taken into consideration.**
>
> Re: We agree that it's crucial to present a more comprehensive view of data consumption, factoring in both image-text pairs and the language data leveraged in P-Former training. While our original rationale was centered around the ease of accessibility and abundance of unimodal language data relative to image-text pairs, we understand the necessity of providing a balanced representation. Additionally, it's essential to underscore that the overhead of training the P-Former, although present, is a one-time process for a given LLM. This means it does not necessitate repeated training for different tasks (e.g., image/video/audio or QA/caption) provided the same OPT-2.7b is utilized. In our revised manuscript, we will ensure that these points are clearly articulated, and the computation overhead section is enhanced.
>
> **W2: After reading the paper, it is still a bit unclear why, in essence, P-Former is effective. Does it function as some type of knowledge distillation? Please provide more discussion about this.**
>
> Re:
> - In our experiments with base-models like BLIP-2, the architecture consists of three sequential components: (1) ViT, (2) VL-connector, and (3) LLM decoder. Given that we use a frozen LLM for generation, optimizing closer to the LLM decoder becomes more pivotal for achieving optimal generation quality.
> - The unique design of P-Former mirrors a sentence embedding model, as evidenced in lines 158 to 163. This means the prompts predicted by the P-Former carry rich semantics. Therefore, during evaluations on unfamiliar images, the model boasts an improved generalization capability.
> - BLIP2's studies indicate that direct end-to-end optimization of the sequential model can sometimes lead to catastrophic forgetting. Our approach adds an additional layer of complexity by decomposing the two-stage BLIP2 training into three stages, further addressing this optimization challenge.
> - For BLIP2, optimization of soft prompt is learned only using text from image-text pair, while our decoupled training allows for leveraging additional unimodal data for optimizing these soft prompts
>
>
> We will include a discussion of this intuition in our revised manuscript.

---

> > ### Comment · Reviewer_T7A1 · 2023-08-18
> >
> > Thanks for the reply. My concerns are basically addressed.

---

### Official Review · Reviewer_8epX · 2023-07-07

**Soundness:** 2 fair
**Presentation:** 2 fair
**Contribution:** 2 fair
**Rating:** 5
**Confidence:** 5

**Summary:**

This paper introduces a prompt-transformer (P-Former), which is pretrained on text corpus, that can trigger the LLM to generate better text prompts to align with the vision-and-language models with better visual features. Empirical experiments on the top of BLIP-2 shows promising results on several tasks.

**Strengths:**

1. This paper introduces an separate stage for P-Former, to trigger LLM to generate text prompts to align with visual features, verifies the ideas on the BLIP-2, and shows promising results on several tasks.

**Weaknesses:**

1. May need more comprehensive experiments to show where is the gain from.

**Questions:**

1. Where is the gain from? I assume the stage 1 in the Figure 3 and Line 211, is just taken from the pretrained BLIP model, right? If not, do you take pretrained BLIP checkpoint for continuing pretrain 10 epochs? And do you have a comparison for the stage 1 pretraining, compared to the BLIP-2? The main issue here is, to try to make an alignment with BLIP-2 to see where the difference/improvement comes from? stage 1 or stage 2.

2. Is the P-Former is agnostic to the different Vision-encoders? if changing a vision encoder, does the framework still work?

**Limitations:**

See the questions part.

---

> ### Author Rebuttal · Authors · 2023-08-10
>
> **W1 and Q1: Where is the gain from? I assume the stage 1 in the Figure 3 and Line 211, is just taken from the pretrained BLIP model, right?**
>
> Re: Our approach in stage 1 is grounded in equation (8) and comprises dual learning objectives: the first one originates from BLIP2, while the second alignment loss is introduced by our P-Former. We have not adopted pretrained BLIP2 weights; we only employ its objective functions. The only learnable parameters in equation (8) are those in Q-Former, which are randomly initialized.
>
> **If not, do you take pretrained BLIP checkpoint for continuing pretrain 10 epochs? And do you have a comparison for the stage 1 pretraining, compared to the BLIP-2? The main issue here is, to try to make an alignment with BLIP-2 to see where the difference/improvement comes from? stage 1 or stage 2.**
>
> Re: Regarding the checkpoint continuation and comparison with BLIP-2: We initiate our vision-to-language model, ViT-Qformer-OPT, with a pretrained ViT and OPT, whereas the Q-former remains randomly initialized. A preliminary "stage 0" serves to train the P-former, which is subsequently frozen. Our subsequent approach can be summarized thusly:
> - Stage 1: We train the ViT (which remains frozen) and the learnable Q-former using equation (8), with the Q-former being initialized randomly.
> - Stage 2: Here, we employ the Q-former from stage 1 and train the ViT-Qfromer-OPT tandem via equation (9).
>
> To succinctly delineate our method from BLIP2: Our stage 1 is represented by equation (8). Our stage 2 is represented by equation (9). By contrast, BLIP2's stage 1 corresponds to equation (8) but with w1 set to 0, and BLIP2's stage 2 matches equation (9) but with w2 set to 0.
>
> **Where does the difference/improvement come from? stage 1 or stage 2.**
>
> Re: By introducing w1 and w2 in equations (8) and (9) respectively, we've enabled an ablation to answer this question. As depicted in our ablation study (Table 4), we examine cases with w1=0 and w2=0. By setting w1 or w2 to zero, we effectively bypass the influence of the P-former, making the approach analogous to the original BLIP-2 training. The results indicate that the alignment loss introduced by our P-former contributes to both stage 1 and stage 2.
>
>
> **Q2: Is the P-Former agnostic to the different Vision-encoders? if changing a vision encoder, does the framework still work?**
>
> Re: Our framework is intrinsically adaptable to different encoders. This is illustrated in section 4.5 where we transition from the attention-driven ViT to a convolution-centric I3D model for video encoding. Notwithstanding this shift, our results (displayed in Table 7) remain very competitive on video-to-language generation.

---

> > ### Comment · Reviewer_8epX · 2023-08-17
> >
> > Thanks for the authors' response. May lend to recommend a weak acceptance, either 5 or 6.

---

### Official Review · Reviewer_jFy5 · 2023-07-09

**Soundness:** 3 good
**Presentation:** 3 good
**Contribution:** 2 fair
**Rating:** 6
**Confidence:** 3

**Summary:**

To improve image-to-text generation, this paper proposes a proxy model P-Former to predict LLM prompts and uses it as an auxiliary loss in BLIP-2 to align selected features with LLM prompts. Results show promising results, especially in 0-shot VQA tasks.


**Strengths:**

Nice novelty by introducing a proxy model for LLM prompts prediction to enhance image-to-text generation.

The experiments, ablations and analysis are comprehensive, showing improvements in both VQA (more significant) and captioning (less significant) tasks, and no improvement in image-text retrieval.

Well written and easy to read.


**Weaknesses:**

This paper only shows the effect of the LLM prompt prediction in an incremental way, i.e., as an auxiliary loss in the existing BLIP-2. It would be more interesting if we could show the effectiveness of P-Former in a cleaner (simpler) setup.

minor: The paper title says “Bootstrapping Vision-Language Learning”, which might be generic according to the results of this paper, such as no improvements on image-text retrieval tasks.


**Questions:**

As mentioned in “Weaknesses”, instead plugging P-Former in BLIP-2 as an auxiliary loss (IMO more like an incremental work), it might be more interesting to verify P-Former’s effect by directly predicting the LLM prompts in a captioning model?

In line 216, also mention the cost to train P-Former?

In line 196, elaborate the size of each dataset?

minor comment: in equation 7, the parentheses seem incorrect?


**Limitations:**

No specific concern.

---

> ### Author Rebuttal · Authors · 2023-08-10
>
> **W1 and Q1: This paper only shows the effect of the LLM prompt prediction in an incremental way, i.e., as an auxiliary loss in the existing BLIP-2. It would be more interesting if we could show the effectiveness of P-Former in a cleaner (simpler) setup.**
>
> Re: A potential pitfall in bypassing the LM loss entirely is that errors originating at the prompt level might be amplified as they navigate through the LM decoder. Given the end goal is caption/text generation, the LM loss remains a pivotal factor for ensuring the proficiency of such a model. Consequently, it may be challenging to completely forgo the LM loss, especially when considering a language generative model.
>
> Further emphasizing the versatility of our method, it's adaptable across various modalities, including image, video, and audio. Moreover, it can be integrated into any VL model that employs prompts as interfaces. To illustrate this adaptability, we reference Section 4.5 where we highlighted the efficacy of our approach in video captioning, **leveraging a model distinct from BLIP2**.
>
> **Q2: In line 216, also mention the cost to train P-Former? In line 196, elaborate the size of each dataset? minor comment: in equation 7, the parentheses seem incorrect?**
>
> Re: These are thoughtful suggestions. We will make the necessary updates in our final version. Thank you!

---

> > ### Comment · Reviewer_jFy5 · 2023-08-17
> >
> > > Given the end goal is caption/text generation, the LM loss remains a pivotal factor for ensuring the proficiency of such a model.
> >
> > IIUC, this is done by the frozen LLM instead of Q-Former?
> >
> > Given the input of the LLM are tokens, can we train a clean generative model to predict these tokens (generated by the pre-trained P-Former). It's fine to reuse Q-Former's architecture and training recipe, but my concern is that why we couldn't use predicting P-Former tokens as the only objective (I assume such a generative model is not that large, unlike the frozen LLM, so maybe this is still doable)?

---

> > > ### Author Response · Authors · 2023-08-18
> > >
> > > Theoretically, it is feasible to train our generative model (ViT+Qformer), denoted as `Gen()`, with the singular objective of predicting the prompt `prompt_gt` generated by the P-Former:
> > > ```
> > > prompt_gen = Gen(Image)
> > > L = (prompt_gen - prompt_gt) ** 2.0
> > > ```
> > > Nevertheless, driving this loss to an exact zero is an unattainable goal. Significantly, **any deviation in learning the prompt is amplified when the prompt is subsequently passed into the LLM** given that generation is our end objective:
> > > ```
> > > caption_gen = LLM(prompt_gen)
> > > ```
> > > Even a minor discrepancy in `prompt_gen` can disproportionately affect the accuracy of `caption_gen`, especially when processed through an intricate LLM.
> > >
> > > The suggestion by the reviewer to rely solely on the objective of prompt alignment loss `L = (prompt_gen - prompt_gt) ** 2.0` is theoretically valid. However, our empirical studies indicate that it fails to deliver the most desirable outcomes:
> > >
> > > Specifically, when testing the alignment loss in isolation (without the LM loss) on the VATEX dataset for video captioning, we observed scores of CIDEr: 35.1, BLEU-4 19.8, and ROUGE: 42.1. These results accentuate that our P-Former can adeptly guide vision encoders along favorable trajectories even in the absence of the LM loss.
> > >
> > > Nevertheless, these scores are inferior to those reported in Table 7. One primary reason is that relying only on the prompt alignment loss excludes both the forward and backward passes in the computationally intensive LLM during training. It's worth noting that the LLM represents over 70% of the parameters in the entire model of BLIP-2 ViT-g OPT2.7B (and surpasses 90% in BLIP-2 ViT-g FlanT5XXL). This not only pertains to a performance-tradeoff but also to the cost of training.
> > >
> > > While our model demonstrates promising results, it cannot replicate these results without the incorporation of the LM loss. Herein, our prompt alignment loss serves as an intermediary, aligning the Q-Former more cohesively with the frozen LLM. We concur with the sentiment that a learning framework that circumvents the LLM, yet maintains robust performance, would indeed be groundbreaking due to the substantial reductions in training duration and computational demands.

---

> > > > ### Comment · Reviewer_jFy5 · 2023-08-18
> > > >
> > > > > L = (prompt_gen - prompt_gt) ** 2.0
> > > >
> > > > I thought the input for the frozen LLM is **tokens**, but from the formula above, we are actually using **embeddings** as the input -- yeah, this would be hard for the Q-Former to learn indeed; also, it seems too difficult or impossible to learn prompt tokens instead of prompt embeddings for P-Former?

---

> > > > > ### Author Response · Authors · 2023-08-19
> > > > >
> > > > > You are right in observing that the input to the LLM is based on soft prompts, which are, in essence, embeddings. To align with terminologies often used in literature, the terms "soft-prompts" and "visual-tokens" sometimes both refer to embeddings, especially in the context of VLM, distinguishing them from traditional text-tokens. Our decision to use soft-prompts in our framework is rooted in two core considerations:
> > > > >
> > > > > - **Compatibility with Established Approaches**: Several significant efforts in the domain, such as BLIP2, Frozen, ClipCap, X-VLLM, and even more recent endeavors utilizing LLaMA, have all adopted **soft-prompts** (embeddings) as the primary input for the LLM. In choosing this path, our objective was to ensure that our methodology was in harmony with current best practices in the field.
> > > > >
> > > > > - **Challenges with Text Tokens**: To the best of our knowledge, there haven't been prominent studies that use text tokens (i.e., discrete integer indices) as prompts for LLM in end-to-end VLM systems. One primary reason for this might be the challenge of differentiability. When utilizing text tokens, one would need to apply an 'argmax' operation on the output from the Q-former (or similar VL connectors) before introducing it to the LLM. Although there are methods like gumbel-softmax, vector quantization, or stop gradients that potentially bypass this obstacle, the intrinsic optimization challenges these present might be why contemporary research consistently veers towards the use of soft-prompts (or embeddings) as the input for LLM.
> > > > >
> > > > > Basically, while the idea of using token indices might seem more straightforward, the complexities it introduces and the desire to maintain consistency with existing works led us to opt for embeddings as the more pragmatic choice for our model.

---

> > > > > > ### Comment · Reviewer_jFy5 · 2023-08-19
> > > > > >
> > > > > > Thanks for the clarification, which has addressed all my concerns!

---

### Official Review · Reviewer_3ni9 · 2023-07-27

**Soundness:** 3 good
**Presentation:** 3 good
**Contribution:** 3 good
**Rating:** 6
**Confidence:** 4

**Summary:**

The paper introduces a novel approach for optimizing the application of large language models in resource-intensive vision-language pre-training. Unlike the traditional approaches of using visual features as prompts to guide language models, the paper focuses on identifying optimal prompts to align with visual features. This is achieved through the introduction of the P-Former, a module trained only on linguistic data, eliminating the need for image-text pairings.

**Strengths:**

1. Originality:
   The paper demonstrates a high level of originality in its approach to vision-language learning. By introducing the P-Former, the paper presents a unique perspective in contrast to the standard approach of using visual features as prompts. Also, the focus on a modality-agnostic framework expands the applicability to various VL tasks, further enhancing the originality of the paper.
2. Quality:
   The paper exhibits a good level of quality in various aspects. The methodology is well-formulated, and the introduction of P-Former is well-justified. Also, it provides sufficient details about the experiments and demonstrates the robustness and flexibility of the framework across different VL tasks.
3. Clarity:
   The paper is written with clarity and precision. The abstract, introduction, and related work sections provide a clear context and properly position the paper in the literature. The technical concepts are well-explained, including the P-Former and the decoupled language pre-training approach. The use of figures and tables further help in understanding the experiments and results.
4. Significance:
   The significance of the paper is in the decoupled language pre-training approach, with the P-Former's ability to predict ideal prompts, addressing resource-intensive challenges in this field. Also, it offers a fresh perspective on how to leverage large LMs in VL learning scenarios.

**Weaknesses:**

- The idea of the P-Former is interesting, however, it seems to lack an intuitive explanation and motivation. Why learning an ideal language prompt helps more, compared to using visual ones as in the counterpart models?
- There seems to be a lack of some ablations, which naturally arise as questions, for example, experimenting with and without the P-Former module, maybe just by using randomly initialized prompt p, using different sizes/types of language backbones, different ways of initializing the prompt p at the beginning, etc.
- Also, there is a lack of qualitative analysis of the experiments. I would recommend including and analyzing qualitative results in comparison to existing approaches. Presenting visual examples of the model's performance in both successful and failure cases can make the paper stronger.
- Some experiments lack stronger interpretation. I would also encourage the authors to provide more interpretation of some results (e.g. tables 1, 2, 3), rather than describing the tables, to enable the readers to gain better insights into the behavior of the framework.

**Questions:**

- Could the authors provide more details and clearer insights into the P-Former part of the framework and size?
- Across all experiments, it can be seen that more data helps BLIP-2 to outperform the P-Former. I wonder if the P-Former will improve the best performance of BLIP-2 if the pre-training image-text data is larger (129M)?
- In Table 3, it can be observed that the performance of the proposed model is lower when it comes to retrieval. Could you interpret these results and elaborate more on this limited performance?
- What is the reasoning behind the need for a separate pre-training of the P-Former on language data? Why do additional unimodal sentences contribute to the performance in Table 6?
- Since the framework is presented as modality agnostic, do you think it can handle language-only tasks, such as machine translation? If so, which parts of the framework would need to be adapted to handle this?

**Limitations:**

The authors have provided a sufficient discussion of the limitations of their work in section 5. Due to the usage of pre-trained backbones, I would also encourage the authors to discuss any potential negative societal impact of their work.

---

> ### Author Rebuttal · Authors · 2023-08-09
>
> **W1: Seems to lack an intuition on why learning an ideal language prompt helps?**
>
> Re:
> - Models like BLIP2 consist of three sequential components: (1) ViT, (2) VL-connector, and (3) LLM decoder. Given that we use a LLM for generation, optimizing closer to the LLM (i.e., prompts, comparing to visual features) becomes more pivotal for achieving optimal generation quality.
> - The design of P-Former mirrors a sentence embedding model (lines 158 to 163). This means the prompts predicted by the P-Former carry rich semantics. Therefore, the model boasts an improved generalization capability.
> - BLIP2's studies indicate direct end-to-end optimization can sometimes lead to catastrophic forgetting. Our approach decomposes the 2-stage training into 3 stages, further addressing this optimization challenge.
> - BLIP2's (implicit) refinement for soft prompts is exclusively achieved through the utilization of textual content from image-text pairs. In contrast, our decoupled training approach empowers us to harness supplementary unimodal data, facilitating the enhancement of these soft prompts in a more comprehensive manner.
>
> **W2: Lack some ablations, e.g., w/o the P-Former module, maybe randomly initialized prompt, different backbones, etc.**
>
> Re: That is a great question which highlights critical facets of the experimentation process.
> - *Random Initialization and Learning Without P-Former*: Our initial approach was, as you mentioned, to directly learn from a randomly initialized prompt p without incorporating the P-former. But, upon testing, we identified a significant challenge. For a smaller model variant like opt-2.7b, which possesses a hidden size of 2560, if we employ 32 tokens as soft prompts for an expansive dataset with 4M sentences, the resultant model would have to accommodate an overwhelming 327B parameters. This would not only have computational implications but also learning from such a vast parameter space can dilute the essential semantic connections between various sentences.
> - *P-Former's Efficiency in Parameterization*: The P-Former emerged as a solution to this parameter explosion problem. P-Former parameterizes the soft prompt $p$ using a semantically-rich model. This design ensures that the total number of parameters remains fixed at 110M. The major advantage here is scalability. Whether we're working with a dataset of 4M, 12M, or 129M, or any LM decoder's size, the P-Former guarantees consistent parameters, making the model more computationally efficient and preventing the loss of essential semantic relationships.
>
> Future iterations of our research will undoubtedly delve deeper into the areas you've highlighted, providing a more comprehensive understanding of the P-Former's capabilities.
>
> **W3: Lack of qualitative analysis**
>
> Re: We value the feedback. We've incorporated qualitative comparisons for datasets such as GQA and OKVQA, allowing us to offer more nuanced insights (in uploaded PDF). We have demonstrated several examples comparing our model's response with BLIP-2 and the ground truth (GT). We will include a section on qualitative analysis in the appendix of our revised paper.
>
> **W4: Provide more interpretation of some results, rather than describing the tables.**
>
> Re: Thank you for the suggestion. We agree that more interpretation of the results will make our paper stronger. We will update our results section to include such a discussion.
>
> **Q1: Provide more details and clearer insights into the P-Former**
>
> Re: This is similar to our response to W1 and W2. Please see our responses above.
>
> **Q2: if the P-Former will improve BLIP-2 in 129M dataset setting?**
>
> Re: Conducting experiments with the 129M dataset presents challenges for us, given that we possess a max of 8 GPUs, while the original results reported by BLIP2 on the 129M dataset utilized 16 GPUs about 9 days. The primary goal of our method is to streamline the training process and make efficient use of the available training data. As such, we anticipate that our approach might show modest gains in a 129M setting, particularly if the model undergoes extensive training. In fact, a key motivation behind P-Former is to reduce the dependence on vast multi-modal datasets and models. This approach not only simplifies the training process but also democratizes participation, ensuring that research in this area isn't solely the domain of entities with access to significant computational resources.
>
> **Q3: Elaborate more on this limited performance in retrieval?**
>
> Re: The retrieval tasks primarily rely on the ViT and Q-former. Since P-former is designed exclusively for language generation by the LLM, it doesn't influence the retrieval processes, especially tasks that rely heavily on the contrastive objective. Our results in Table 3 aim to show that introducing P-Former doesn't negatively affect existing functionalities.
>
> **Q4: reasoning for a separate pre-training of the PFormer on language? Why do additional sentences contribute?**
>
> Re: (1) We drew inspiration from BLIP2’s observations that a 2-stage training is easier to optimize than end-to-end. Our strategy introduces an additional pre-training stage for the P-former, and our results confirm its benefits. (2) Introducing additional unimodal sentences enhances the learning of equation 4. With a fixed language decoder $D$, additional text $t$ inputs facilitate the learning of a better P-former $E$.
>
> **Q5: Language-only tasks, e.g., machine translation?**
>
> Re: Absolutely, our framework can be adapted for language-only tasks, including machine translation. The key adjustments would involve simply substituting the ViT with a language encoder and swapping the VL connector for a language-to-language connector. While a few adjustments are necessary, translations have proven effective when utilizing an encoder coupled with cross-attention, as exemplified in *'Attention is All You Need'*. For tasks focused solely on language, cross-attention might be the more intuitive approach.

---

> > ### Comment · Reviewer_3ni9 · 2023-08-21
> >
> > Thanks for the response, it clarified all my concerns. I would recommend acceptance.

---

> > > ### Author Response · Authors · 2023-08-21
> > > **Thank you!**
> > >
> > > We greatly appreciate the reviewer's confidence in our work.

---

### Author Rebuttal · Authors · 2023-08-10

Dear Reviewers:

We sincerely thank you for your thoughtful reviews and constructive feedback on our paper. We are heartened to see that our proposal of adding a new component, a P-Former, to the X-language (X being any modality) pre-training framework resonated well with all of you. Your appreciation of our exploratory direction, the improvements shown in our experiments, and the potential impact on future research is highly encouraging. Your insights have aided in refining our understanding, and we have addressed each of your comments individually.

In addition to our detailed response, we've incorporated qualitative comparisons of our method and BLIP-2 for specific datasets, such as GQA and OKVQA (see attached pdf). We will include a section on qualitative analysis in the appendix of our revised paper.

---

### Decision · Program_Chairs · 2023-09-21

**Decision:**

Accept (spotlight)

**Comment:**

This paper proposes a new approach for vision-language learning, in particular visual language models. To this end, the paper introduces a new transformer that generates prompts that optimally align with the visual features. The reviewers highlighted the papers originality, experimental completeness, unique ideas and relevance to the community. In addition, the extensive rebuttal and discussion have further cleared *all* of the remaining questions. Recommending spotlight.